# Increasing the precision of orthology-based complex prediction through network alignment

Roland A. Pache[1,3] and Patrick Aloy[1,2]

[1] Joint IRB-BSC Program in Computational Biology, Institute for Research in Biomedicine (IRB Barcelona), Barcelona, Spain
[2] Institució Catalana de Recerca i Estudis Avançats (ICREA), Barcelona, Spain
[3] Current affiliation: Department of Bioengineering and Therapeutic Sciences, University of California San Francisco (UCSF), San Francisco, CA, USA

## ABSTRACT

Macromolecular assemblies play an important role in almost all cellular processes. However, despite several large-scale studies, our current knowledge about protein complexes is still quite limited, thus advocating the use of *in silico* predictions to gather information on complex composition in model organisms. Since protein–protein interactions present certain constraints on the functional divergence of macromolecular assemblies during evolution, it is possible to predict complexes based on orthology data. Here, we show that incorporating interaction information through network alignment significantly increases the precision of orthology-based complex prediction. Moreover, we performed a large-scale *in silico* screen for protein complexes in human, yeast and fly, through the alignment of hundreds of known complexes to whole organism interactomes. Systematic comparison of the resulting network alignments to all complexes currently known in those species revealed many conserved complexes, as well as several novel complex components. In addition to validating our predictions using orthogonal data, we were able to assign specific functional roles to the predicted complexes. In several cases, the incorporation of interaction data through network alignment allowed to distinguish real complex components from other orthologous proteins. Our analyses indicate that current knowledge of yeast protein complexes exceeds that in other organisms and that predicting complexes in fly based on human and yeast data is complementary rather than redundant. Lastly, assessing the conservation of protein complexes of the human pathogen *Mycoplasma pneumoniae*, we discovered that its complexes repertoire is different from that of eukaryotes, suggesting new points of therapeutic intervention, whereas targeting the pathogen's Restriction enzyme complex might lead to adverse effects due to its similarity to ATP-dependent metalloproteases in the human host.

Corresponding author
Roland A. Pache,
roland.pache@ucsf.edu

## INTRODUCTION

Almost every major process in a cell, such as replication, transcription, translation and degradation, is carried out not by single proteins, but by macromolecular complexes, regulated through intricate networks of protein–protein interactions. To understand cellular behaviour on a systemic level, we thus need a comprehensive knowledge of the protein complexes present in the respective organism.

In the last years, many small-scale studies have identified protein complexes in yeast and human, which have been collected in the public databases MPACT (*Güldener et al., 2006*) and CORUM (*Ruepp et al., 2010*), respectively. Moreover, several large-scale proteomics experiments, using tandem-affinity purification coupled to mass spectrometry analysis, have focused on systematically unveiling the composition of macromolecular complexes in the budding yeast *Saccharomyces cerevisiae* (*Gavin et al., 2002*; *Ho et al., 2002*; *Gavin et al., 2006*; *Krogan et al., 2006*; *Babu et al., 2012*), the human pathogen *Mycoplasma pneumoniae* (*Kühner et al., 2009*), and recently also in the fruit fly *Drosophila melanogaster* (*Guruharsha et al., 2011*) and in human (*Hutchins et al., 2010*; *Havugimana et al., 2012*). Although yeast is the least complex eukaryotic model organism with about 6,000 genes, several hundreds of protein complexes were found, and the total number of complexes in yeast was estimated to be over 800 (*Gavin et al., 2006*). Moreover, the first screen in *M. pneumoniae* already yielded 62 homo- and 116 heteromultimeric protein complexes, despite the pathogen's small genome of only 689 protein-coding genes (*Kühner et al., 2009*). The importance of protein complexes for cell survival becomes apparent when probing the essentiality of their protein-coding genes through knock-out mutations. Indeed, several studies have shown that protein complexes in yeast are significantly enriched in essential genes (*Dezso, Oltvai & Barabási, 2003*; *Hart, Lee & Marcotte, 2007*; *Wang et al., 2009*; *Pache, Babu & Aloy, 2009*). To discover the molecular details of how individual proteins function together as macromolecular assemblies, follow-up initiatives have then aimed at identifying those complexes that are suitable for structural studies by combining systematic bioinformatics and experimental validation strategies (*Pache & Aloy, 2008*; *Brooks et al., 2010*).

So far, however, these important investigations, which have improved our understanding of the architecture and function of protein complexes, were limited to yeast, not only due to the scarcity of functional data in other organisms, but also because their protein complexes are yet largely unknown. To determine whether the findings for yeast complexes describe general principles of molecular organization and to discover how protein complexes have evolved, it is thus necessary to define protein complexes in other species, complementing the results of recent screening efforts (*Guruharsha et al., 2011*; *Hutchins et al., 2010*; *Havugimana et al., 2012*). Since the experimental characterization of macromolecular assemblies is difficult and requires large amounts of time and resources, predicting protein complexes based on existing protein–protein interaction (PPI) and orthology data becomes an interesting alternative.

Indeed, different strategies have been developed to exploit these data. On the one hand, several graph-clustering strategies have been applied to interactome networks in order to identify functional modules and protein complexes as densely connected subgraphs

(*Spirin & Mirny, 2003*; *Pereira-Leal, Enright & Ouzounis, 2004*; *Poyatos & Hurst, 2004*). In addition, new algorithms have specifically been designed for this task with the aim to distinguish real complex components from spurious interactors and to allow shared components across different complexes (*Bader & Hogue, 2003*; *Pu et al., 2007*). Various clustering techniques were also used to define protein complexes from purification data in the original large-scale screens of macromolecular assemblies in yeast, fly, human and *M. pneumoniae* (*Gavin et al., 2006*; *Krogan et al., 2006*; *Babu et al., 2012*; *Kühner et al., 2009*; *Hutchins et al., 2010*; *Havugimana et al., 2012*). On the other hand, since protein complexes are often conserved due to the constraints PPIs pose on functional divergence during evolution (*Roguev et al., 2008*; *van Dam & Snel, 2008*), it is possible to predict complexes using orthology information. In its simplest form, orthology-based complex prediction reports the collection of all orthologous proteins of a given complex in one species as the corresponding complex in the other organism (*Koonin, Wolf & Aravind, 2001*). However, one-to-many and many-to-many orthology relationships between species often imply functional divergence of paralogous genes after duplication, leading to the prediction of many false complex components with increasing proteome size.

The recent advent of tools for the comparison and alignment of protein interaction networks (*Kelley et al., 2003*; *Sharan et al., 2005*; *Koyutürk et al., 2006*; *Sharan & Ideker, 2006*; *Cootes, Muggleton & Sternberg, 2007*; *Kiemer & Cesareni, 2007*; *Narayanan & Karp, 2007*) now opens up new possibilities for complex prediction. One strategy is to align whole interactome networks of different species to search for conserved functional modules (*Sharan et al., 2005*; *Narayanan & Karp, 2007*; *Hirsh & Sharan, 2007*; *Ali & Deane, 2009*). For instance, *Sharan et al. (2005)* aligned the yeast and *H. pylori* interactomes, finding 11 conserved protein complexes, while Hirsh and colleagues found 150 conserved complexes by aligning the yeast and fly interactomes, matching known complexes in yeast with coherent functional annotations (*Hirsh & Sharan, 2007*). However, interactome to interactome alignment does not exploit knowledge about the composition of known complexes. This can only be done through complex to interactome alignment, in which the network representation of a known query complex in a given organism such as yeast is aligned to the interactome of a target species. For instance, *Dost et al. (2008)* developed the QNet algorithm, which allows the querying of input graphs of treelike topology in interaction networks, and used it to align 94 manually curated yeast complexes from the MPACT database (*Güldener et al., 2006*) to the fly interactome, finding 36 of them to be conserved in fly.

For this work, we applied our recently developed tool for network comparison, NetAligner, which was demonstrated to outperform the current standard in the field in a variety of different benchmarks (*Pache & Aloy, 2012*) and was already used successfully to discover the role of structural disorder in the rewiring of interactomes during evolution (*Mosca, Pache & Aloy, 2012*). Using NetAligner, we investigated how the incorporation of interaction data through network alignment influences the performance of orthology-based complex prediction. Moreover, we systematically aligned known protein complexes in *H. sapiens*, *S. cerevisiae* and *M. pneumoniae* to whole species interactomes to find novel

complex components in yeast and human, predict yet undiscovered complexes in the fly *D. melanogaster* and search for similarities and differences between the complexes repertoires of *H. sapiens* and the human pathogen *M. pneumoniae*.

## RESULTS AND DISCUSSION

### Network alignment increases precision in orthology-based complex prediction

A standard, straightforward method to predict protein complexes in a target species based on those known in a given query organism is the so-called 'orthologs approach'. In that method, the union of all orthologous proteins of the respective query complex components is predicted to constitute that complex in the target species (*Koonin, Wolf & Aravind, 2001*). The general idea behind this approach is that many protein complexes are evolutionarily conserved, because they perform critical cellular tasks, such as replication, transcription or translation, needed in all forms of cellular life. However, due to evolutionary divergence of proteins after duplication, which can lead to functionally non-overlapping paralogs, not all orthologs of the components of a given complex in one species should be expected to be part of the corresponding complex in another organism. Prediction of protein complexes using the standard orthologs approach can thus result in false complex components (i.e., false positive predictions). To test whether the incorporation of interaction information through network alignment can decrease the number of false positives and thus increase the precision of orthology-based complex prediction, we compared the performance of the orthologs approach to that of NetAligner (*Pache & Aloy, 2012*) in predicting yeast protein complexes based on human complexes and vice versa through complex to interactome alignment (Fig. 1A). For this, we used the non-redundant benchmark set of 71 matching human-yeast complex pairs (see *Materials & Methods*), which we previously defined (*Pache & Aloy, 2012*), and analysed the complex predictions of the orthologs approach with respect to how well they agree with the corresponding benchmark set complexes (see *Materials & Methods*). We evaluated both precision (i.e., fraction of true complex components among all proteins predicted to be part of the given complex) and recall (i.e., fraction of complex components recovered in the given prediction) in the same way as in our previous study (*Pache & Aloy, 2012*), so that we could directly compare the performance of the orthologs approach to the performance reported for NetAligner in predicting protein complexes via complex to interactome alignment (see below and *Materials & Methods*). We found that incorporating interaction data through network alignment significantly increases the precision of orthology-based complex prediction (i.e., the ability to distinguish orthologs that are part of the complex in the target species from those that are not). When using NetAligner with default parameters there is a significant increase in precision from 34.6% to 54.1% ($p$-value $= 2.04 \times 10^{-32}$, one-sided Fisher's exact test), which is also present when calibrating NetAligner for the alignment of human complexes to the yeast interactome (53.9%, $p$-value $= 4.07 \times 10^{-32}$, one-sided Fisher's exact test) or vice versa (58.5%, $p$-value $= 2.65 \times 10^{-36}$, one-sided Fisher's exact test; Fig. 1B). This increase in precision

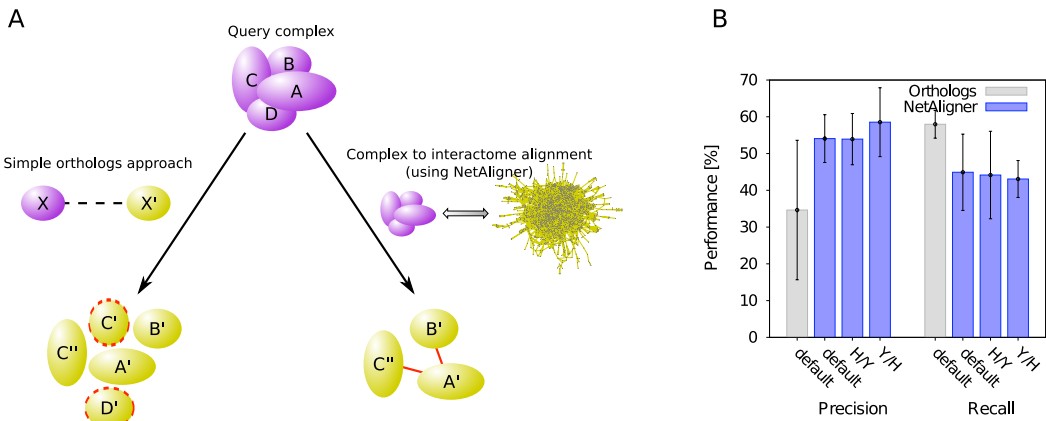

**Figure 1 Network alignment vs. simple orthologs approach in predicting protein complexes.** Comparison of network alignment (using NetAligner *Pache & Aloy, 2012*) and a simple orthologs approach in predicting protein complexes in a target species based on known complexes in a query organism. (A) In the simple orthologs approach, all proteins in the target species (yellow) that are orthologous to the proteins of the query complex (violet) are predicted to form the given complex in the target organism. This can lead to false positives (proteins marked with red dashed lines) that are not part of the real complex. In the more sophisticated network alignment strategy, only those orthologs are predicted to be part of the given complex in the target species that are supported by conserved or likely-conserved interactions (red) between the two organism interactomes. (B) Performance of the orthologs (grey) and NetAligner (blue) methods on a benchmark set of known conserved human/yeast complex pairs, using default parameters (default) or parameter sets trained on one species pair (H/Y, Y/H) and evaluated across both species pairs. Error bars denote one standard error of the mean.

arises from NetAligner using protein–protein interaction data to identify and filter out false positive orthology mappings. Due to current interactome networks still being incomplete and missing many interactions existing in nature (*Venkatesan et al., 2009*), this filtering step unavoidably leads to an increase in false negatives and thus a reduced recall of true complex components (from 58.0% to 44.9% with default parameters; Fig. 1B). This is caused by orthologous proteins getting filtered out of complex predictions, since they seem not to interact with any other complex component, while those interactions indeed do exist and are just missing from current interactome networks. So currently, there is a tradeoff between precision and recall when using NetAligner to predict protein complexes. However, once species interactomes become more complete, we would expect network alignment approaches to deliver increased precision at the same level of recall of simple orthology mapping. Ongoing and future interaction detection experiments should increase interactome coverage and thus also increase the recall of true complex components when predicting protein complexes through network alignment.

## Strategy to predict complexes through complex to interactome alignment

To predict complexes in yeast, human and fly, which are the species with the most interaction data available, and to identify novel complex components, we aligned the non-redundant sets of human, yeast and mycoplasma protein complexes (see *Materials & Methods*) to whole species interactomes using NetAligner (*Pache & Aloy, 2012*). In

**Table 1 NetAligner parameters and expected alignment performance.** The NetAligner parameters for complex to interactome alignment were taken from *Pache & Aloy (2012)*. For yeast to human and human to yeast alignments, we took the best performing parameter combinations for these species as determined in the benchmarks (*Pache & Aloy, 2012*). For all other species, we used the default parameters for complex to interactome alignment. The option to predict likely conserved interactions is always set to true (1), because this considerably improves alignment performance (*Pache & Aloy, 2012*). Precision and recall describe the expected performance of NetAligner in correctly identifying protein complex components (see Materials & Methods).

| | | Complex to interactome alignment | | |
|---|---|---|---|---|
| | | **Yeast to human** | **Human to yeast** | **Other species** |
| **Parameters** | Predict likely conserved interactions | 1 | 1 | 1 |
| | Vertex probability threshold | 0.0 | 0.0 | 0.0 |
| | Edge probability threshold | 0.1 | 0.0 | 0.0 |
| | Max insertion length | 1 | 2 | 2 |
| | Vertex to edge score balance | 0.2 | 0.0 | 0.1 |
| **Performance** | Precision [%] | 49.17 | 60.91 | 54.07 |
| | Recall [%] | 38.06 | 56.06 | 44.91 |

contrast to a pure orthologs-approach, network alignment via NetAligner incorporates knowledge about protein–protein interactions into orthology-based complex prediction. When aligning a query complex to the interactome of a target species, NetAligner aligns those pairs of orthologous proteins that are part of the two input networks and identifies conserved and likely conserved interactions, as well as parts where the query complex and target interactome differ slightly, represented through gaps and mismatches in the alignment graph constructed by the program (*Pache & Aloy, 2012*). Several program parameters, such as the vertex and edge probability thresholds, further determine which pairs of orthologous proteins will be part of the final alignment solutions. For those alignment scenarios that were part of the complex to interactome alignment benchmark reported in *Pache & Aloy (2012)* (i.e., yeast to human and human to yeast), we could use the best-performing parameter combinations, while for the others there is no benchmark data available, and we thus used the default parameters (Table 1). We considered only the highest-ranked significant alignment solution (with a standard *p*-value threshold of 0.05) for each query complex.

## Finding novel components of yeast and human protein complexes

To find novel components of protein complexes in yeast and human, we aligned the non-redundant sets of 1027 human complexes and 244 yeast complexes from the manually curated databases CORUM (*Ruepp et al., 2010*) and MPACT (*Güldener et al., 2006*), to the yeast and human interactome, respectively (see *Materials & Methods*). Here, we did not use the complexes identified in large-scale studies to ensure that our predictions are based only on curated data sources. This yielded 257 non-redundant significant complex predictions in yeast and 89 in human (Table 2 and Figs. 2A and 2C). We then identified novel components by comparing our complex predictions to all known complexes in

**Table 2 Complex predictions in yeast and human.** Basic statistics of the complex prediction results in yeast and human, based on aligning known human complexes to the yeast interactome and vice versa (see Materials & Methods). Results are shown both for all complex predictions (All) and for the high-confidence subset (HC). #, number of; nr, non-redundant.

| Prediction type | Human to yeast | | Yeast to human | |
|---|---|---|---|---|
| | **All** | **HC** | **All** | **HC** |
| # query complexes | 1027 | 645 | 244 | 236 |
| # predicted complexes (nr) | 257 | 105 | 89 | 61 |
| Total # complex components (nr) | 604 | 372 | 464 | 325 |
| Average # proteins per complex | 6.4 | 5.93 | 6.37 | 5.67 |
| Total # novel components (nr) | 307 | 107 | 175 | 91 |
| Average # novel components per complex | 2.75 | 1.46 | 2.4 | 1.62 |
| # entirely novel complexes (nr) | 0 | 0 | 2 | 2 |

the respective species (see *Materials & Methods*). We found 307 non-redundant novel components across 181 yeast complexes and 175 non-redundant novel components in 65 human complexes (Table 2 and Figs. 2B and 2D). Given the recall of the method when aligning human complexes to the yeast interactome and vice versa (Table 1), most of the predicted complexes are probably sub-complexes (as our method misses some complex components due to incomplete interaction data). Moreover, based on the precision of our method (Table 1), we can estimate that at least 60% of the novel yeast complex components we predicted (from aligning human complexes to the yeast interactome) and 49% of the novel human components we predicted (from aligning known yeast complexes to the human interactome) are real complex members. For an independent *in silico* validation of our predictions, we computed the number of predicted complexes that are functionally homogeneous (see *Materials & Methods*) separately for each Gene Ontology (GO) category (i.e., biological process, molecular function and cellular component) (*Ashburner et al., 2000*), and compared it to the respective number of functionally homogeneous query complexes and predictions based on simple orthology mapping (Figs. 3A and 3B). We found that, even when requiring the complexes to fulfill this criterion in at least two GO categories (e.g., 'biological process' and 'molecular function'; see *Materials & Methods*), the majority of the predicted complexes in yeast (70%) and human (73%) are indeed functionally homogeneous. The fact that the fraction of human query complexes that are homogeneous is smaller than the fraction of predicted yeast complexes (63% vs. 70%) might indicate that our predictions are, on average, of similar quality as the manually-curated human complexes stored in the CORUM database (*Ruepp et al., 2010*). When using a simple orthologs approach instead of NetAligner, we got a similar fraction of homogeneous complex predictions in yeast (73%). On the other hand, the fraction of homogeneous yeast query complexes is larger than the fraction of homogeneous complex predictions in human (97% vs. 73%), suggesting that in this case, our predictions are of lower quality than the query complexes. However, when using a simple orthologs approach, the resulting fraction of homogeneous complex predictions in human is even
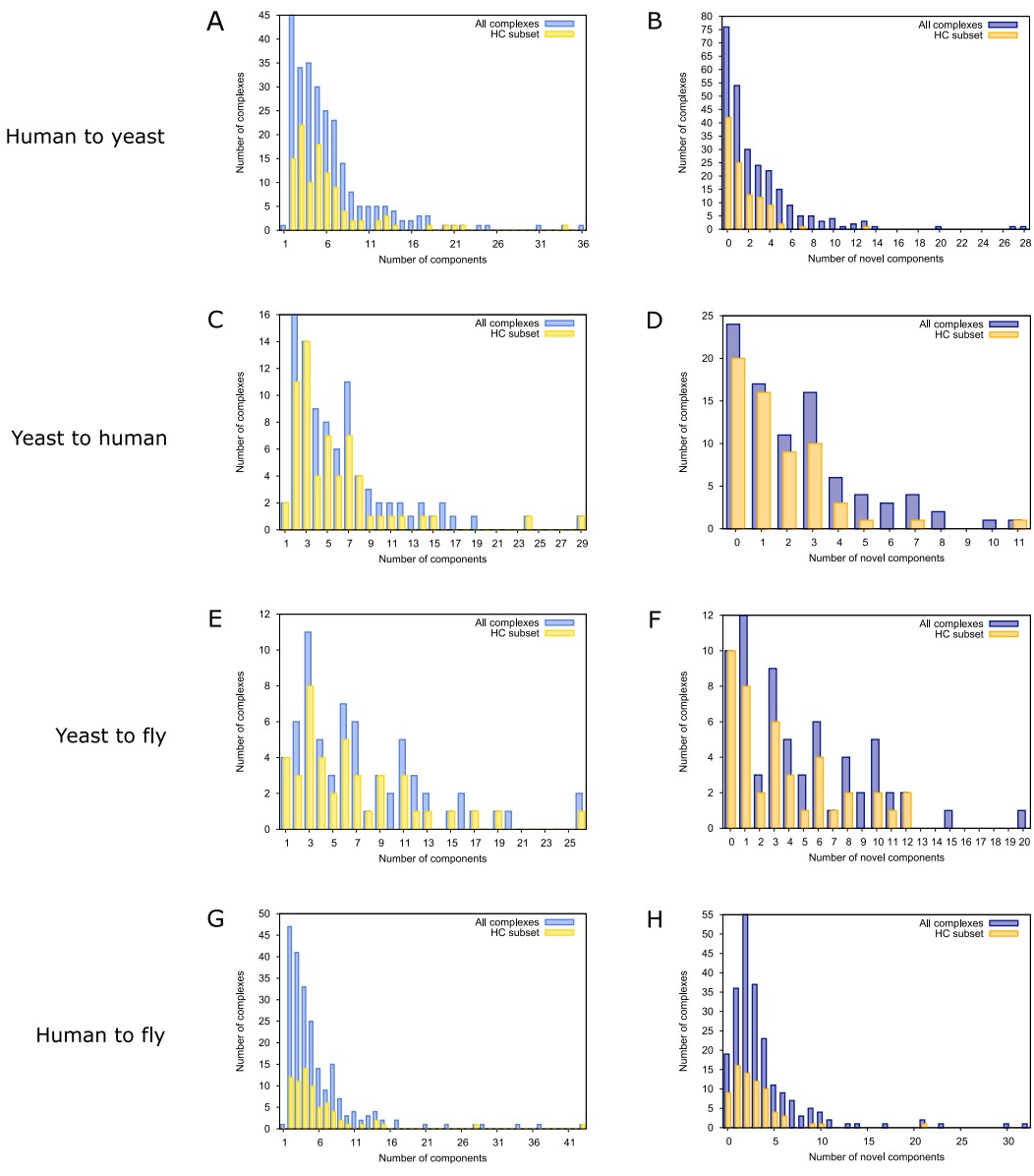

**Figure 2 Complex size and number of novel components distributions for all predicted complexes.** Distributions of the total number of components of all predicted complexes (blue) and the high-confidence (HC) subset (yellow) are shown on the left. Distributions of the number of novel components found in all predicted complexes (violet) and in the HC subset (orange) are shown on the right. (A) & (B) prediction of yeast complexes based on human data; (C) & (D) prediction of human complexes based on yeast data; (E) & (F) prediction of fly complexes based on yeast data; (G) & (H) prediction of fly complexes based on human data.

lower with only 58% (Figs. 3A and 3B). A likely explanation for these results is that our current knowledge about protein complexes is considerably better in yeast than in human, with several real human complex components still missing the respective functional annotations and thus leading to less homogeneous complex predictions. Nevertheless, in most cases we tested, using NetAligner led to a higher fraction of homogeneous complex

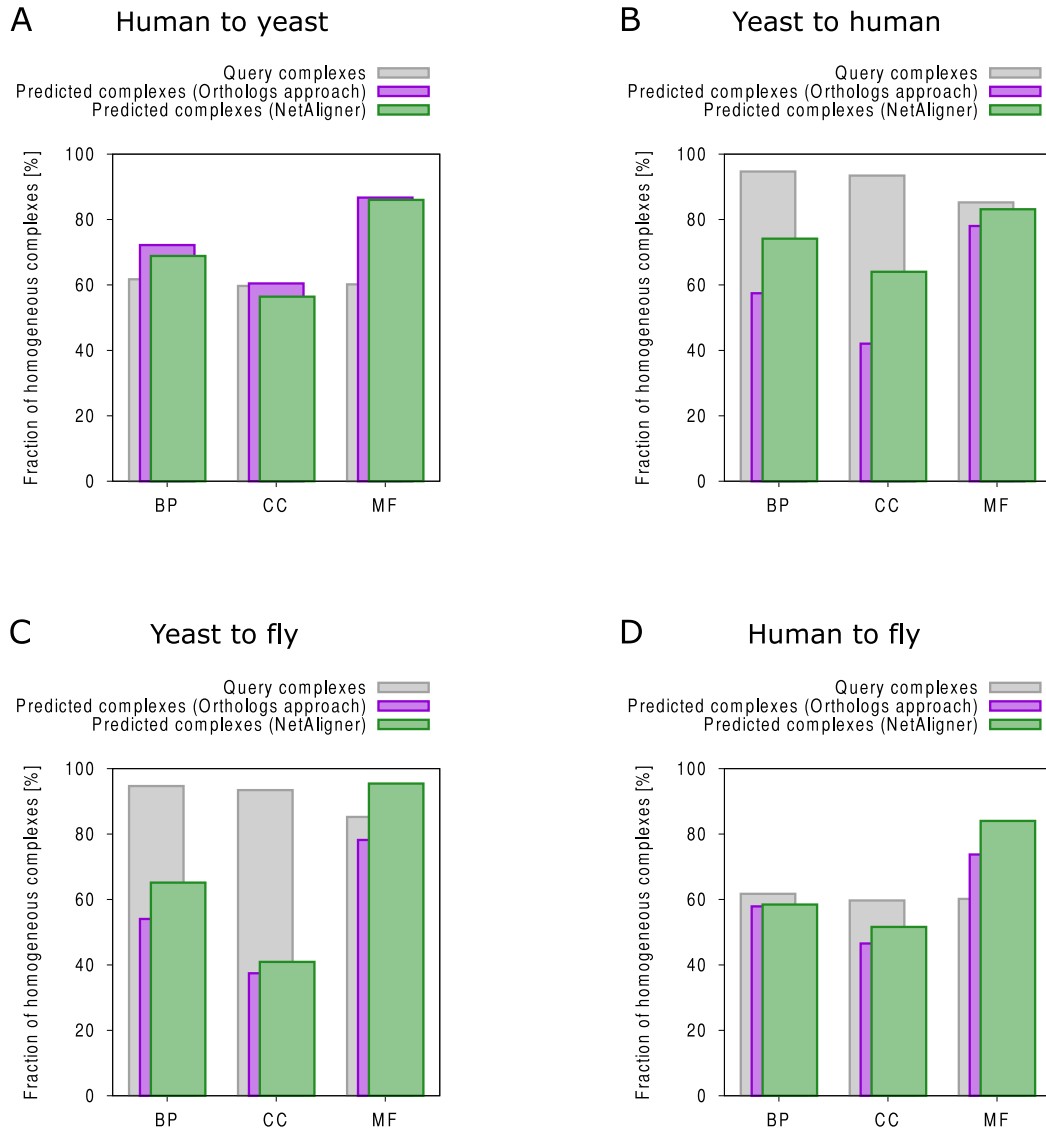

**Figure 3 Functional homogeneity of query and predicted complexes.** Comparison of the functional homogeneity of query (grey) and predicted complexes, using either a simple orthologs approach (purple) or NetAligner (green), in the different Gene Ontology (GO) annotation categories. BP, Biological Process; CC, Cellular Component; MF, Molecular Function. (A) prediction of yeast complexes based on human data; (B) prediction of human complexes based on yeast data; (C) prediction of fly complexes based on yeast data; (D) prediction of fly complexes based on human data.

predictions than a simple orthologs approach (Fig. 3). In addition, we tested whether the annotations of the predicted complexes hint towards specific functional roles in the cell by computing the statistical significance of GO term enrichments in each complex (see *Materials & Methods*). We found that 233 of our complex predictions in yeast (91%) and 87 in human (98%) were significantly functionally enriched with respect to the given species interactome, indeed suggesting specific cellular roles for those complexes and further validating our predictions. For the complete list of predicted complexes,

see Table S1, in which we annotated the complexes with the GO terms that are shared by most complex components to provide information about their possible biological roles and highlighted those functional annotations that are significantly enriched in the given complex. We also created subsets of high-confidence (HC) predictions (Table S1), requiring each member complex to be functionally homogeneous and originate from a homogeneous query complex with which it shares at least one homogeneous GO term (see *Materials & Methods*). This resulted in HC sets of 105 yeast and 61 human complex predictions (Table 2 and Figs. 2A and 2C), with 98% and 97% of them, respectively, being significantly functionally enriched with respect to the given species interactome. We found 107 non-redundant novel components across 63 HC yeast complexes and 91 non-redundant novel components in 41 HC human complexes (Table 2 and Figs. 2B and 2D). For instance, we predicted the proteins MPPA, MPPB, QCR1 and QCR2 to form a complex in human (Fig. 4A). All four proteins are orthologous to the alpha and beta subunits of the Mitochondrial processing peptidase (MPP) complex in yeast, which is involved in the maturation of mitochondrial proteins by proteolytic cleavage of the N-terminal localization sequence (*Nomura et al., 2006*). NetAligner found the interaction between MPPA and MPPB of the yeast query complex to be conserved between human MPPA and MPPB, as well as between the QCR1 and QCR2 orthologs (also known as UCR-1 and UCR-2). In addition, this interaction was predicted to be likely conserved between the other components of the complex (Fig. 4A), but no subcomplex of any of the four proteins was found in current databases. According to our *in silico* validation experiments, all components of the predicted complex are involved in proteolysis and have metalloendopeptidase activity, but two of them (MPPA and MPPB) localise to the mitochondrial matrix, while the other two (QCR1 and QCR2) localise to the mitochondrial inner membrane as core components of the Cytochrome bc1 complex. Although it might be that the MPP and QCR subunits form two separate complexes *in vivo*, a combined MPP/QCR complex might also exist, since the two subcellular localizations are adjacent, and it was observed that in plants, the MPP complex is actually integrated into the Cytochrome bc1 complex, with QCR1 and QCR2 being identical to MPPB and MPPA, respectively (*Nomura et al., 2006*). Another interesting example is the alignment of the human EXO1-MLH1-PCNA complex, which is involved in DNA-mismatch repair, to the yeast interactome (Fig. 4B). The yeast complex predicted by the alignment solution consists of six different proteins, PCNA, RAD27, DIN7, EXO1, MLH1 and MLH3, based on interactions existing either in human or yeast and predicted to be likely conserved in the other species (Fig. 4B). The DNA sliding clamp PCNA and the endonuclease RAD27 (also known as FEN1) are known to form a complex in DNA replication and repair (*Gomes & Burgers, 2000*), and the interaction between MLH1 and MLH3 plays an important role in meiotic recombination and mismatch repair (*Wang, Kleckner & Hunter, 1999*). In addition, the double-stranded DNA exonucleases EXO1 and DIN7, both participating in mismatch repair, have high sequence similarity, and the double knockout of EXO1 and RAD27 is lethal (*Tishkoff et al., 1997*). Together, these findings point towards the possibility of a

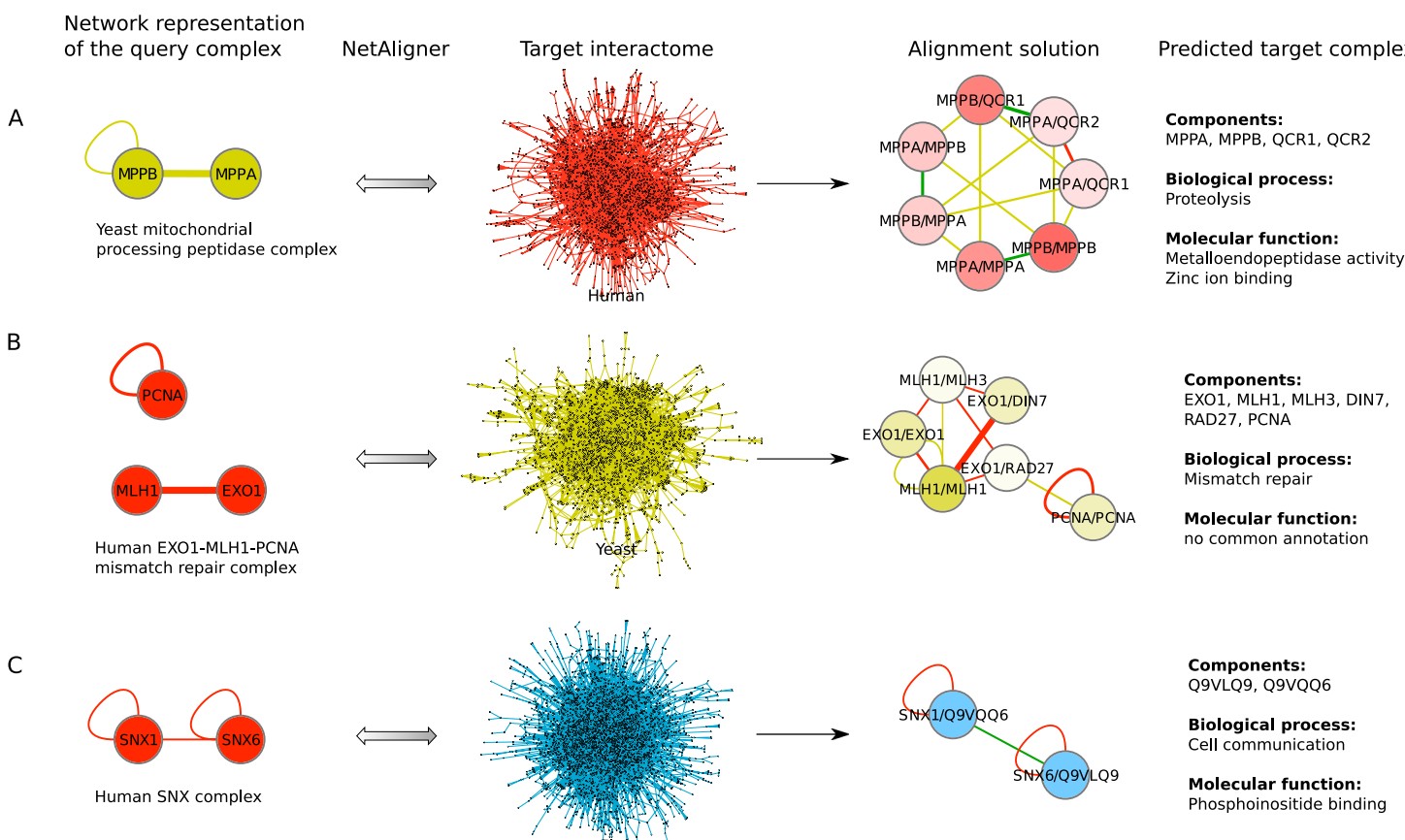

**Figure 4  Examples for complex prediction based on network alignment.** Examples of using NetAligner (*Pache & Aloy, 2012*) to predict protein complexes based on aligning network representations of known protein complexes in one species to the interactome of another species. Notably, NetAligner does not require query complexes to be fully connected. Edge colors in the alignment solutions show which species the given interaction that was predicted to be likely conserved originates from. Green edges denote known conserved interactions. High-confidence interactions are highlighted with thicker edges. Alignment solution nodes represent pairs of orthologous proteins (separated by '/') between the respective species, allowing both one-to-many and many-to-many orthology relationships. Components of the given predicted complex in the target species (extracted from the respective alignment solution) are listed, together with the GO biological process and molecular function annotations that could be assigned to the predicted complex. (A) prediction of a mitochondrial processing peptidase complex in human; (B) prediction of a mismatch repair complex in yeast; (C) prediction of a SNX complex in fly. See main text for details. Network representations were created with Cytoscape (*Smoot et al., 2011*).

six-component mismatch-repair complex in yeast and also that the corresponding human complex might have additional components.

Finally, comparing the distributions of the total number of predicted (Figs. 2A and 2C) and novel components (Figs. 2B and 2D), we found that both achieve higher values in yeast than in human. However, this likely originates from the set of human query complexes simply being considerably larger (1027 complexes) than the set of yeast query complexes (244 complexes) and thus leading to both more complex predictions and a higher total number of novel components in yeast. On the other hand, the exponential decrease in the number of novel yeast complex components we observed (Fig. 2B) compared to the considerably broader distribution of novel human complex components (Fig. 2D), supports the view that our knowledge about yeast complexes surpasses that of human

**Table 3 Prediction of protein complexes in fly.** Basic statistics of the complex prediction results in fly, based on aligning known yeast and human complexes to the fly interactome (see Materials & Methods). Results are shown both for all complex predictions (All) and for the high-confidence subset (HC). #, number of; nr, non-redundant.

| Prediction type | Yeast to fly | | Human to fly | |
|---|---|---|---|---|
| | All | HC | All | HC |
| # query complexes | 244 | 236 | 1027 | 645 |
| # predicted complexes (nr) | 66 | 42 | 219 | 71 |
| Total # complex components (nr) | 405 | 255 | 640 | 291 |
| Average # proteins per complex | 7.5 | 6.69 | 5.84 | 5.93 |
| Total # novel components (nr) | 252 | 134 | 454 | 168 |
| Average # novel components per complex | 4.61 | 3.55 | 3.69 | 2.79 |
| # entirely novel complexes (nr) | 6 | 4 | 45 | 11 |

ones. This is because, relative to complex predictions in yeast, a larger fraction of predicted human complexes contains a given number of novel components. For instance, only 24 out of 257 predicted yeast complexes (9.3%) contain three novel components, while the same is true for 16 out of 89 predicted human complexes (18%).

## Predicting fly complexes from human and yeast data is complementary rather than redundant

The fruitfly *Drosophila melanogaster* is an important model organism. However, there does not yet exist a dedicated database of protein complexes in fly, the first large-scale screen, reporting 556 complexes, has only recently been completed (*Guruharsha et al., 2011*), and only 221 complexes are annotated in GO (*Ashburner et al., 2000*). We thus predicted protein complexes in that species by aligning the non-redundant sets of 244 yeast complexes and 1027 human complexes to the fly interactome (see *Materials & Methods*). This resulted in 66 non-redundant significant complex predictions originating from yeast and 219 from human (Table 3 and Figs. 2E and 2G), with only little overlap (see below). Compared to the set of 777 known complexes in fly (based on GO annotations and the recent large-scale screen by *Guruharsha et al. (2011)*; see *Materials & Methods*), our complex predictions contained 252 non-redundant novel components across 56 complexes based on yeast data and 454 non-redundant novel components in 200 complexes originating from human data (Table 3 and Figs. 2F and 2H). Among those complexes, 6 and 45, respectively, were entirely novel. Again based on the recall and precision of the method (Table 1), we can estimate that most of the predicted complexes are probably sub-complexes and that at least 54% of the novel components we found are real complex members. The independent *in silico* validation (see *Materials & Methods*) revealed that the majority of the predicted fly complexes are functionally homogeneous, independent of whether they originated from yeast (70%) or human (68%) query complexes (Figs. 3C and 3D). This indicates that both of those organisms represent interesting sources for predicting evolutionary conserved protein complexes in fly based on network alignment. The fraction of homogeneous predicted fly complexes also lies between that of the human

(63%) and yeast (97%) query complexes, indicating that our predictions are, on average, of at least the same quality as the manually-curated human complexes stored in the CORUM database (*Ruepp et al., 2010*), but of lower quality than the manually-curated yeast complexes in MPACT (*Güldener et al., 2006*). Here, we observed the biggest loss of homogeneity in the cellular component category (Fig. 3C). This might, however, result from the respective fly proteins missing sub-cellular annotations, which would again suggest that the current knowledge about protein complexes in yeast surpasses that of complexes in other species. Using a simple orthologs approach instead of NetAligner again leads to lower fractions of homogeneous complex predictions, independent of whether they originated from yeast (58%) or human (60%) query complexes. Among the fly complex predictions, 58 based on yeast (88%) and 189 based on human data (86%) were significantly functionally enriched with respect to the fly interactome (see *Materials & Methods*), indicating specific cellular roles for those complexes and further validating our predictions. For the complete list of all predicted fly complexes, see Table S2. The HC subsets of fly complex predictions (see *Materials & Methods* and Table S2) consist of 42 fly complexes originating from yeast and 71 from human data (Table 3 and Figs. 2E and 2G), with 95% and 94% of them, respectively, being significantly functionally enriched with respect to the fly interactome. We found 134 non-redundant novel components across 32 HC complexes predicted from yeast and 168 non-redundant novel components in 62 HC complexes originating from human query complexes, with 4 and 11 of those complexes being completely novel (Table 3 and Figs. 2F and 2H). For instance, aligning the yeast Replication factor C (RFC) complex, consisting of the components RFC1-5, to the fly interactome predicted the corresponding fly assembly to encompass six proteins, Q9VKW3, Q9VX15, RFC1, RFC2, Q7KLW6 and Q9U9Q1 (Fig. S1). According to the complex annotations in GO (*Ashburner et al., 2000*), the first four of those are known to belong to the RFC complex in fly, while the latter two are novel. Our alignments showed that both Q7KLW6 and Q9U9Q1 are orthologous to RFC2-5 in yeast, and our *in silico* validation experiments revealed that all six predicted components are involved in DNA replication and have ATP-binding capability, important for performing the loading of the DNA sliding clamp (*Gomes, Schmidt & Burgers, 2001*; *Schmidt, Gomes & Burgers, 2001*). Moreover, in the HTP screen of *Guruharsha et al. (2011)*, the first five of those proteins were purified together, which provides further evidence for Q7KLW6 to actually be a component of the RFC complex in fly. On the other hand, this is also an example where NetAligner (through the incorporation of interaction data) was able to filter out false positive components that a simple orthologs approach would have predicted to belong to the complex. These comprise the proteins Q8T3K3, Q8IQ05 and Q95WV5, which are annotated as DNA replication accessory factors and most similar to the yeast chromosome transmission fidelity protein 18 (CTF18), which is known to substitute RFC1 in the RFC-like complex that also contains the proteins CTF8 and DCC1 and is required for establishment of chromosome cohesion in the S-phase of the cell cycle (*Mayer et al., 2001*; *Naiki et al., 2001*). Moreover, the SNX subcomplex of the human Retromer complex, which is involved in mediating endosome to trans-Golgi network retrograde transport
(*Wassmer et al., 2007*; *Hong et al., 2009*), represents an example for predicting fly complexes based on human data (Fig. 4C). It consists of the SNX1-SNX6 dimer that is important for membrane-bound coat formation (*Wassmer et al., 2007*; *Hong et al., 2009*), and we predict this complex to exist in fly as an assembly of Q9VQQ6 and Q9VLQ9. This prediction is not only supported by SNX1 and SNX6 being the best BLAST (*Altschul et al., 1997*) hits of Q9VQQ6 and Q9VLQ9, respectively, but also through the interaction between the two human components being conserved in fly (Fig. 4C). Together, the examples shown in Fig. 4 illustrate the fact that query complexes need not be fully connected (i.e., can contain isolated proteins), but rather that NetAligner is indeed capable of identifying conserved protein complexes despite the incompleteness of current interactome networks (*Venkatesan et al., 2009*).

Comparing the fly complex predictions from yeast and human data (Fig. 5), we found only 26 pairs of matching complexes (i.e., pairs of predicted fly complexes with a component overlap of more than 50% of each complex), covering 21 non-redundant predicted complexes from human (10%) and 12 from yeast data (18%). One reason for this little overlap between the fly complex predictions is probably the low number of matching human and yeast query complexes. Indeed, only 41 (4%) and 46 (19%) of the non-redundant sets of 1027 human and 244 yeast query complexes, respectively, are present in the non-redundant set of matching human/yeast complex pairs. Between the HC subsets, there are only 11 pairs of matching complexes, covering 11 non-redundant predicted HC complexes from human (15%) and 7 from yeast data (17%; Fig. 5). Moreover, none of the completely novel complexes we predicted was found based on both yeast and human data. This clearly indicates that predicting fly complexes from yeast and human query complexes through network alignment is complementary rather than redundant. Protein complexes that were found both when starting from human and from yeast data include the Replication factor C complex, the Casein kinase II, the 20S core and 19/22S regulatory particles of the proteasome, as well as the Septin, Tubulin and Actin filament complexes (Fig. 5), which all represent well-studied conserved assemblies.

## Mycoplasma complexes differ substantially from those of eukaryotes

*Kühner et al. (2009)* reported the first genome-wide analysis of protein complexes in the human pathogen *Mycoplasma pneumoniae*, which has one of the smallest known genomes (689 protein-encoding genes). This analysis revealed a rather complicated machinery of almost 200 protein complexes, of which the majority were yet unknown (*Kühner et al., 2009*). To predict whether some of these are actually conserved in other organisms, we aligned the non-redundant set of 174 mycoplasma complexes to the interactomes of yeast, fly and human (see *Materials & Methods*). The complex to interactome alignments led to only 11, 9 and 6 non-redundant significant predictions in those species, respectively (Table 4). Compared to the sets of known protein complexes (see *Materials & Methods*), our predictions contained 86 non-redundant novel components across 6 yeast complexes, 68 across 9 fly complexes and 30 in 5 human complexes (Table 4). Based on the recall

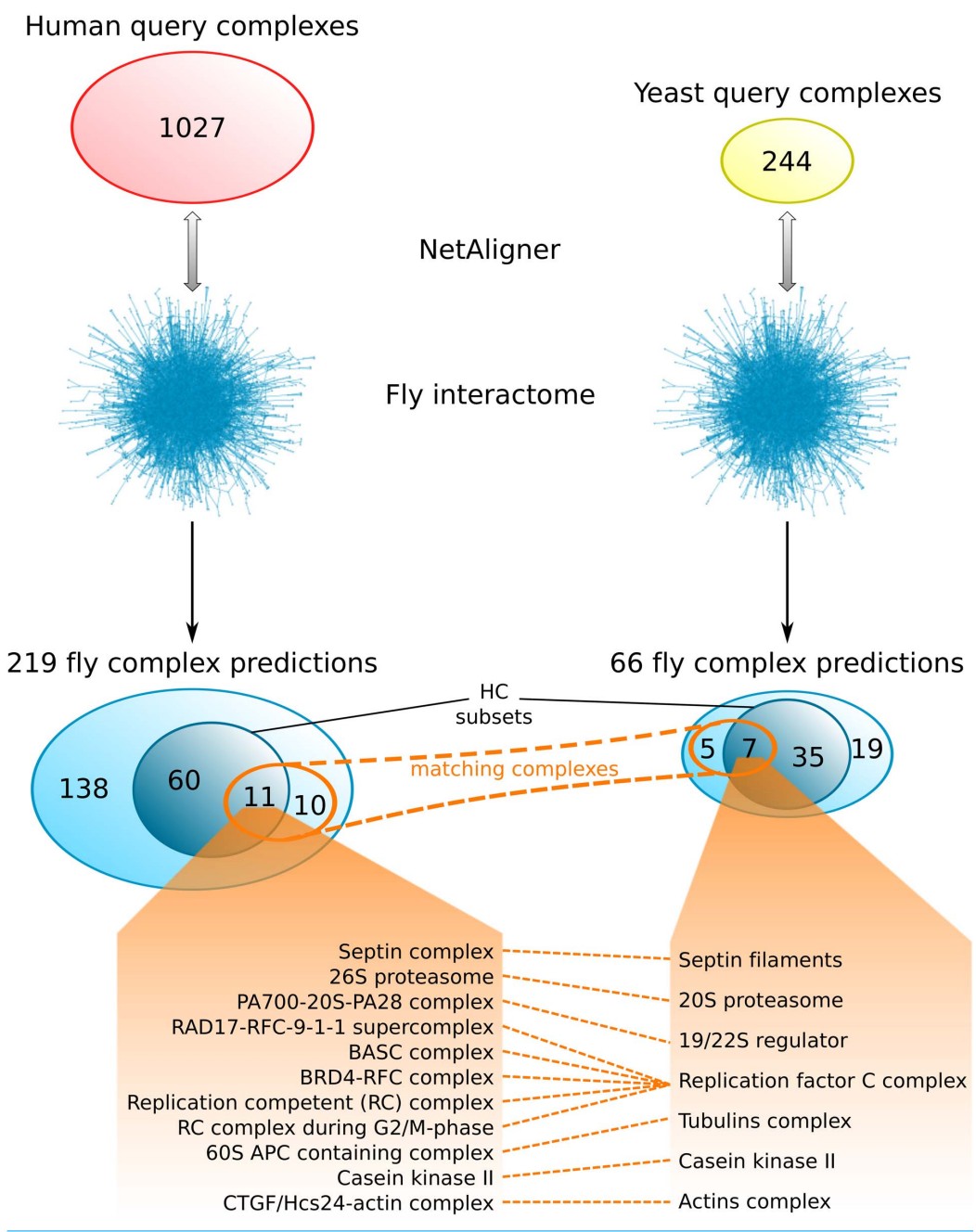

**Figure 5 Comparison of fly complex predictions originating from yeast and human data.** Comparison of the 219 and 66 non-redundant fly complexes predicted through alignment of human (red) and yeast (yellow) query complexes, respectively, to the fly interactome using NetAligner (*Pache & Aloy, 2012*). High-confidence (HC) subsets of the complex predictions are shown in darker blue. Matching complexes, i.e., those that were identified both from human and yeast query complexes with more than 50% shared components, are highlighted in orange. The 11 pairs of matching complexes between the two HC subsets are listed below.

**Table 4 Predictions based on known mycoplasma complexes.** Basic statistics of the complex prediction results in yeast, fly and human, based on aligning known mycoplasma complexes to the respective species interactome (see Materials & Methods). Results are shown both for all complex predictions (All) and for the high-confidence subset (HC). #, number of; nr, non-redundant.

| | Mycoplasma to | | | | | |
| | Yeast | | Fly | | Human | |
| Prediction type | All | HC | All | HC | All | HC |
|---|---|---|---|---|---|---|
| # query complexes | 174 | 67 | 174 | 67 | 174 | 67 |
| # predicted complexes (nr) | 11 | 3 | 9 | 1 | 6 | 1 |
| Total # complex components (nr) | 125 | 8 | 86 | 6 | 39 | 3 |
| Average # components per complex | 13.64 | 2.67 | 12.0 | 6 | 7.17 | 3 |
| Total # novel components (nr) | 86 | 1 | 68 | 4 | 30 | 0 |
| Average # novel components per complex | 9.36 | 0.33 | 9.56 | 4 | 5.33 | 0 |
| # entirely novel complexes (nr) | 0 | 0 | 0 | 0 | 1 | 0 |

and precision of the method (Table 1), we can again estimate that most of the predicted complexes are probably sub-complexes and that at least 54% of the novel components we found are real complex members. According to our independent *in silico* validation (see *Materials & Methods*), only 55%, 44% and 33% of all predicted yeast, fly and human complexes, respectively, are functionally homogeneous. If the mycoplasma interactome (on which the query complexes are based) contained many false positive interactions, one potential reason for the observed low functional homogeneity could be that these complex predictions are less reliable. Other possible reasons include inherent differences in the complexes repertoires of those species or aspects of biology that are less well studied. In contrast, the fact that only 39% of all query mycoplasma complexes are homogeneous is likely due to a lack of functional annotations in that organism.

Overall, since we used only the high-confidence subset of mycoplasma interactions (to reduce the number of false positives), we think the very low numbers of significant complex predictions (independent of the target species) and their low functional homogeneity suggest that the mycoplasma interactome (at least the part currently known) is very different from the interactomes of yeast, fly and human. This indicates that there probably exist protein complexes that are unique to the pathogen and might thus be targeted by drugs without causing adverse effects in the human host.

All complex predictions, except for one in yeast, were significantly functionally enriched with respect to the given species interactome, suggesting that they perform specific biological roles. For the complete list of predicted yeast, fly and human complexes, based on mycoplasma data, see Table S3. The HC subsets of those complex predictions (see *Materials & Methods* and Table S3), consist of only three predicted complexes in yeast (one novel component), one in fly (four novel components) and one in human (no novel components; Table 4), with all of them being significantly functionally enriched with respect to the given species interactome. The mycoplasma query complexes that were predicted to be conserved in those HC sets comprise the DNA polymerase III complex in

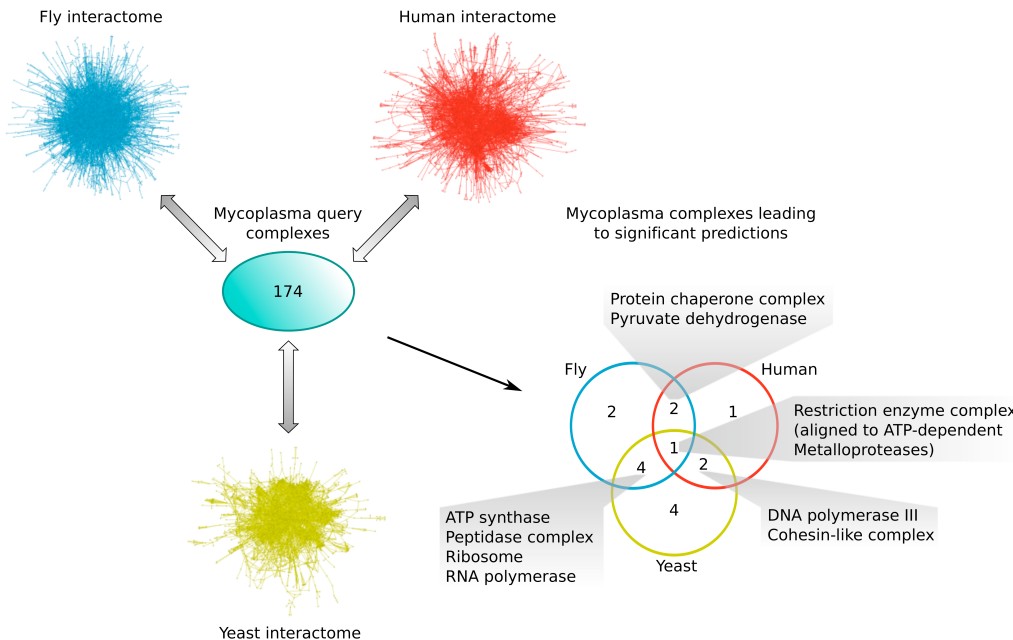

**Figure 6  Mycoplasma complexes leading to significant predictions in yeast, fly and human.** Comparison of the different sets of mycoplasma complexes that led to significant predictions in yeast (yellow), fly (blue) and human (red) through complex to network alignment using NetAligner (*Pache & Aloy, 2012*) (grey arrows). Overlaps between the different sets are shown as a Venn diagram. Complexes found in at least two species are listed.

human, the ATP synthase complex in fly and yeast, as well as the 6-Phosphofructokinase and the Ribonucleoside-diphosphate reductase in yeast. The DNA polymerase III complex, consisting of DPO3X and the yet uncharacterized proteins Y007 and Y450, represents an interesting case: it was aligned to the RFC complex in human, whose clamp loading function is actually incorporated into the DNA polymerase III holoenzyme complex in prokaryotic species such as mycoplasma (*Pomerantz & O'Donnell, 2007*).

Comparing the different subsets of mycoplasma complexes that led to significant predictions in the other species (Fig. 6), one complex, the Restriction enzyme complex, could be aligned to protein complexes in all three species interactomes, two additional complexes were found in both yeast and human (the DNA polymerase III and Cohesin-like complexes), two in both fly and human (the Protein chaperone and Pyruvate dehydrogenase complexes), and four complexes were found in both the yeast and fly interactome (the ATP synthase, Peptidase, Ribosome and RNA polymerase complexes; Fig. 6). This means that the majority of mycoplasma complexes that led to significant predictions (56%) could be found in more than one target species, representing complexes involved in core biological processes conserved from bacteria up to higher eukaryotes. The alignment of the Restriction enzyme complex was, at first, surprising, because it is known to exist only in bacteria and archaea. A closer inspection, however, revealed that it was aligned to ATP-dependent metalloproteases in the eukaryotic species, suggesting that targeting the

Restriction enzyme complex with drugs might also affect ATP-dependent metalloproteases in the human host and thus lead to potential adverse effects.

## CONCLUSIONS

Protein complexes represent key molecular entities that are implicated in many important biological processes within a cell. However, complexes are yet largely uncharacterized in most species and experimental determination of their composition is still a costly endeavour. Increasing our knowledge about protein complexes in important model organisms via complex prediction thus represents an attractive option. Here, we showed that incorporating PPI data through network alignment significantly increases the precision of orthology-based complex prediction, though at the expense of missing some real complex components. By aligning known protein complexes to the interactomes of human, yeast and fly (the species with the highest number of interactions currently available in public databases) using NetAligner (*Pache & Aloy, 2012*), our recently developed tool for network alignment, we were able to identify conserved protein (sub)complexes between human and yeast, as well as novel complex components, with higher precision than by using a simple orthologs-based approach. In addition, we predicted novel macromolecular assemblies (not present in current databases) in fly based on known yeast and human complexes, and found that our contemporary knowledge of yeast complexes surpasses that of other species, which is not surprising given the huge efforts invested into detecting macromolecular assemblies in yeast (*Gavin et al., 2002*; *Ho et al., 2002*; *Gavin et al., 2006*; *Krogan et al., 2006*; *Babu et al., 2012*). On the other hand, we found that current human and yeast complex data leads to complementary predictions in fly, meaning that querying known macromolecular assemblies of those species in the fly interactome unveils different conserved complexes. In the *in silico* validation experiments that we performed, our NetAligner-based complex predictions had about the same functional homogeneity as the known complexes we used for querying, and we were able to assign specific functional roles to almost all complexes. In contrast, predictions based on simple orthology mapping often resulted in reduced functional homogeneity. Finally, aligning the protein complexes of the human pathogen *M. pneumoniae* (*Kühner et al., 2009*) to the interactomes of human, yeast and fly yielded only a handful of significant complex predictions, suggesting that the mycoplasma interactome is at least in parts quite different from those of eukaryotic species. Quantifying those differences is very difficult due to the incompleteness of current interactome data (*Venkatesan et al., 2009*). Nevertheless, since this issue is at least partly addressed through the prediction of likely conserved interactions by the NetAligner algorithm, it indicates that some of the protein complexes that have been identified in mycoplasma might not have any counterpart in eukaryotes and thus represent interesting drug targets with little chances of causing adverse effects in the human host. On the other hand, our analyses revealed that the pathogen's Restriction enzyme complex, which is known to be unique to bacteria and archaea, is similar to ATP-dependent metalloproteases in human, implying that targeting this complex might have undesirable outcomes.

## MATERIALS & METHODS

### Datasets of protein sequences

We collected datasets of protein sequences for human (*Homo sapiens*), fly (*Drosophila melanogaster*), yeast (*Saccharomyces cerevisiae*) and mycoplasma (*Mycoplasma pneumoniae*) from the UniProt Knowledgebase release 15.8 (*UniProt-Consortium, 2009*) by merging the set of sequences stored in Swiss-Prot (including splice variants) and TrEMBL with experimental evidence on protein or transcript level. For mycoplasma, due to a lack of annotation data, we also included sequences not yet marked as having experimental evidence, but excluded all sequences that were only present in TrEMBL and thus of low confidence. Clustering based on 100% sequence identity resulted in non-redundant sets of 75,981 human, 23,296 fly, 6,121 yeast and 687 mycoplasma protein sequences.

### Lists of orthologous proteins

We determined lists of orthologous proteins for species combinations of human, fly, yeast and mycoplasma by performing a reciprocal BLASTP (*Altschul et al., 1997*) search. We used an E-value threshold of $10^{-10}$ and considered only hits in the top10 of the BLASTP output to filter out spurious matches. This resulted in non-redundant sets of 91,112 human/fly, 19,558 human/yeast, 12,778 fly/yeast, 1,005 human/mycoplasma, 644 fly/mycoplasma and 488 yeast/mycoplasma orthologs.

### Interactome construction

We built interactome networks for human, fly and yeast by extracting and merging binary protein–protein interaction data from the major interaction databases IntAct (*Aranda et al., 2010*), MINT (*Ceol et al., 2010*) and HPRD (for human) (*Keshava Prasad et al., 2009*). To increase the quality of the resulting binary interactome networks, we then filtered out all those interactions without support in form of Pubmed ID(s) or interaction detection method(s). For mycoplasma, we extracted the list of high-confidence, binary protein–protein interactions from *Kühner et al. (2009)*, mapping ordered locus names to UniProt accession codes (*UniProt-Consortium, 2009*). This resulted in non-redundant interactomes consisting of 53,290 interactions in human, 19,260 in fly, 60,721 in yeast and 1,058 in mycoplasma (Table S4).

### Non-redundant benchmark set of human/yeast complex pairs

We used the non-redundant benchmark set of conserved human/yeast complex pairs described in *Pache & Aloy (2012)*. In brief, this set is based on the manually-curated yeast complexes from the MPACT database (*Güldener et al., 2006*) and human complexes from the CORUM database (*Ruepp et al., 2010*) whose components are fully present in the respective species interactome, and clustered to remove redundancy. Matching (i.e., conserved) complexes between the two species are defined by requiring at least 2 and 25% of the components of the given human complex to have at least one ortholog in the respective yeast complex and vice versa. The complete benchmark set encompasses 71 matching human/yeast complex pairs, consisting of 64 non-redundant human and 52 non-redundant yeast complexes.

## Complex to interactome alignment using NetAligner

We performed all complex to interactome alignments using the NetAligner algorithm that we recently developed (*Pache & Aloy, 2012*) and which is also available as a web server (*Pache, Céol & Aloy, 2012*). For all species combinations, we computed vertex and interaction conservation probabilities required by NetAligner based on the interactomes and lists of orthologous proteins as described in *Pache & Aloy (2012)*, using default parameters. We assigned reliabilities to each interaction based on the number of Pubmed IDs supporting it as previously described (*Kelley et al., 2003*). For aligning the complexes to whole organism interactomes, we created a network representation of each complex, consisting of all interactions between complex components that are present in the respective species interactome and including self-interactions with a reliability of 0 for all singletons in order to not lose any information about complex composition (*Pache & Aloy, 2012*).

## Performance comparison to simple orthologs approach

To compare the performance of NetAligner to that of a simple orthologs approach, in which the set of all orthologs of the components of a given query complex are predicted to be part of the complex in the target species, we evaluated the results of this approach with respect to the benchmark set of protein complexes as described in *Pache & Aloy (2012)*. In brief, the complex predictions are evaluated in terms of how well they agree with the respective matching benchmark set complexes based on the overlap of their protein components. For each complex predicted by the orthologs approach, we first determined the best-matching benchmark complex of the same species by minimising the total number of unmatched components. A complex prediction was deemed to 'cover' a benchmark complex if it contained at least 2 and at least 50% of its components. We then calculated the number of true positives (TP) as the total number of distinct proteins common to any given complex prediction and the benchmark complex it covers; the number of false positives (FP) as the total number of distinct proteins that are part of any given complex prediction, but not present in the benchmark complex it covers; and the number of false negatives (FN) as the total number of distinct proteins present in any given benchmark complex, but not part of any complex prediction covering that complex. Next, we computed the performance of the orthologs approach in terms of precision and recall:

$$\text{precision} = \frac{\text{TP}}{\text{TP} + \text{FP}}; \quad \text{recall} = \frac{\text{TP}}{\text{TP} + \text{FN}}.$$

Finally, we report the average precision and recall of predicting yeast complexes based on human protein complex data and vice versa to avoid parameter overfitting (Fig. 1B). For NetAligner, since we used the same performance evaluation strategy, as well as the same list of orthologs and set of benchmark complexes, we could directly take the performance results reported in our previous work for complex to interactome alignment (*Pache & Aloy, 2012*). We also evaluated the performance when using default parameters (Fig. 1B).

## Non-redundant sets of protein complexes in human, yeast and mycoplasma

We collected non-redundant sets of protein complexes in human, yeast and mycoplasma. For this, we first extracted all human complexes from the CORUM database (*Ruepp et al., 2010*), the set of manually curated yeast complexes from the MPACT database (*Güldener et al., 2006*), as well as all homo- and heteromeric mycoplasma complexes from *Kühner et al. (2009)*. Analogously to the procedure for constructing the non-redundant benchmark set of human/yeast complex pairs (*Pache & Aloy, 2012*), we then filtered out those complexes that were not fully present in the respective species interactome and clustered them based on the overlap of their components using complete linkage hierarchical clustering to remove redundancy. The distance $d(c_1, c_2)$ between two complexes $c_1$ and $c_2$ was defined as:

$$d(c_1, c_2) = 1 - \frac{|c_1 \cap c_2|}{\max(|c_1|, |c_2|)}$$

and we cut the resulting dendrogram using a distance threshold of 0.5, such that each pair of complexes in the same cluster share more than 50% of their components (choosing the largest complex of each cluster as its representative). This resulted in non-redundant sets of 1027 protein complexes in human, 244 in yeast and 174 in mycoplasma.

## Identification of novel complex components

To identify novel components in our complex predictions, we compared them with the set of known complexes of the respective species. We took all 1826 known human complexes from the CORUM database (*Ruepp et al., 2010*), all 402 human complexes annotated in the Gene Ontology (GO) (*Ashburner et al., 2000*), as well as the 155 and 622 complexes from the recent high-throughput (HTP) studies of *Hutchins et al. (2010)* and *Havugimana et al. (2012)*, respectively (3005 complexes in total). For yeast, we collected all 263 manually curated and all 871 HTP complexes from the MPACT database (*Güldener et al., 2006*), which include the large-scale studies performed by *Gavin et al. (2002)* and *Ho et al. (2002)*. We then added the 491 and 547 complexes defined in the HTP studies by *Gavin et al. (2006)* and *Krogan et al. (2006)*, respectively, as well as all 350 yeast complexes annotated in GO and the recently published set of 720 yeast complexes from *Babu et al. (2012)* (3242 complexes in total). Since there does not yet exist a dedicated database of protein complexes in fly, we determined the set of known fly complexes based on the set of HTP complexes reported by *Guruharsha et al. (2011)* and shared GO annotations (child terms of 'macromolecular complex' (GO:0032991)), similar to *Bruckner et al. (2010)*, resulting in a total of 777 fly protein complexes. For each predicted complex in a given organism, we then determined the known complex of that species with the largest overlap in terms of protein components and marked all those proteins as novel that are not part of the known complex.

## Computation of functional homogeneity

We computed the functional homogeneity of protein complexes as an automated strategy to validate their composition in terms of protein components. First, we extracted the

GO protein annotations from the UniProt database (*UniProt-Consortium, 2009*) for all GO categories (i.e., biological process, molecular function and cellular component). Then, we calculated the GO homogeneity $h(c)$ of each complex $c$ per GO category, defined as the maximum fraction of protein components $p(c)$ that share the same GO term $t$ (*Goh et al., 2007*):

$$h(c) = \max_t \frac{|p_t(c)|}{|p(c)|}.$$

We classified each complex with a GO homogeneity of higher than 0.5 as functionally homogeneous in the given GO category. Lastly, to increase the confidence level of all subsequent analyses, we defined all those complexes as functionally homogeneous that fulfilled this criterion in at least two GO categories.

## Statistical significance of functional enrichments

For all most-abundant functional annotations of a given complex and GO category (i.e., those that contribute to its functional homogeneity in that category), we determined the statistical significance of their enrichments in the complex with respect to the given species interactome, using a one-sided Fisher's exact test with Bonferroni multiple testing correction and a strict $p$-value threshold of 0.025. We then defined all those complexes as significantly functionally enriched that had a significant enrichment $p$-value in at least two GO categories.

## High-confidence subsets of complex predictions

We post-processed all sets of predicted complexes to define high-confidence (HC) subsets. A predicted complex has to fulfil the following criteria to be present in the given HC set: it has to (i) be part of the non-redundant subset of significant predictions, (ii) originate from a functionally homogeneous query complex, (iii) be homogeneous itself and (iv) share at least one homogeneous GO term with the given query complex. Query complexes were defined as high-confidence if they were functionally homogeneous.

## ACKNOWLEDGEMENTS

We would like to thank Sebastian Kühner (Anne-Claude Gavin's lab at EMBL Heidelberg) for providing the mycoplasma complexes and interaction data, as well as Amelie Stein (UCSF) and Andreas Zanzoni (University of Aix-Marseille) for helpful discussions.

### Funding

This work was partially supported by the Spanish Ministerio de Ciencia e Innovación (PSE-010000-2009-1; BIO2010-22073). RAP is a recipient of the Spanish FPU fellowship. The funders had no role in study design, data collection and analysis, decision to publish, or preparation of the manuscript.

## Grant Disclosures

The following grant information was disclosed by the authors:

Spanish Ministerio de Ciencia e Innovación: PSE-010000-2009-1, BIO2010-22073.

## Competing Interests

The authors declare there are no competing interests.

## Author Contributions

- Roland A. Pache conceived and designed the experiments, performed the experiments, analyzed the data, wrote the paper, prepared figures and/or tables, reviewed drafts of the paper.
- Patrick Aloy conceived and designed the experiments, analyzed the data, reviewed drafts of the paper.

## Supplemental Information

Supplemental information for this article can be found online at http://dx.doi.org/10.7717/peerj.413.

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
