# Peer review of "Increasing the precision of orthology-based complex prediction through network alignment"

_PeerJ, doi:10.7717/peerj.413_

## Round 0.1 · original submission · Major Revisions

· Academic Editor

Major Revisions

Both reviewers appreciated your overall approach but had specific comments and concerns. In particular, Reviewer 2 argued that you haven't convincingly shown that NetAligner performs better than orthology mapping. In a revised manuscript, you should strengthen the evidence for your claim (as outlined by the reviewer). If this evidence cannot be provided, then alternatively you will have to weaken your claim.

Note that reviewer 2 alerted me of a typo in the review. In the second paragraph of the section "Validity of the findings", the first sentence should start with "in order to demonstrate", rather than "in addition to demonstrate".

·

Basic reporting

Looks good. No major comments.

Minor comment:

1. Fig 3 and tables: "Human vs Yeast”, "Yeast vs Human" are unclear titles. An example way to improve this would be, rather than using "Human vs Yeast" to indicate yeast predictions based on human input, say “Human into Yeast” or “Yeast from Human”. I’d recommend adopting this throughout the paper, including table headings for example and figure legends. If this is too onerous, instead it could be clearly stated in each figure legend at some point that the first species in _ vs _ is the source/query, the second is the target/predictions.

Experimental design

One significant comment.

1. Line 441 describes the construction and resulting size of the different species interactomes. I believe there is not enough information presented here to replicate the interactome maps they used, which are not made available with the publication. Without experience in building an interactome for these purposed, I tried downloading all human interactions from the IntAct database, removing self-interactions, and keeping only the pairs where both proteins begin with “uniprotkb”. I get 85,206 non-redundant interactions using these steps, using only the IntAct database, so it appears there are filters applied that weren’t stated to result in a merged non-redundant set of around 53,000 interactions.

Validity of the findings

A minor comment.

Line 180: I think the conclusion from these results is stated too strongly. Their line 185 statement seems to me to be broadly accepted—yeast is much more comprehensively functionally annotated than human. It’s good that they mention this alternative explanation, but this explanation seems to be so likely that it’s hard to justify the strong presentation of their first interpretation.

A few significant comments.

1. Figure 1B: Data currently shows a tradeoff between precision and recall for the “simple orthologs” approach vs their NetAligner approach, which doesn’t make it clear to me that this more complex approach is generally better. These could plausibly be different points on the same precision-recall curve. As both approaches seem to result in simple in versus out predictions, as opposed to prediction ranks/probabilities, it may be difficult at present to generate a precision-recall curve. If there was a way to generate ranked predictions from one or the other approach, a precision-recall curve could be generated for at least that approach, answering this concern. Alternatively, might they be able to generate at least a few points on a curve by optimizing the parameters of NetAligner for a few different levels of desired precision or recall? Also, why on line 128 is the lower recall of their approach described as “expected”? Could they dig into this more and explain why their approach isn’t capable of, for instance, improving precision at the same recall level, by tweaking parameters of their method? Could they loosen some criteria of their new approach to include all the predictions made by the simple orthologs approach, adjusted by the information from the alignment? These last steps are not suggested as required, just ideas.

2. Line 169: How are these lower bounds (61% and 50% respectively) on the true positive rate of novel predicted yeast and human complex components arrived at? I don’t see where exactly they follow from Table 1. Maybe just a few words of additional explanation would fix this.

3. Line 242: I’m not sure if I can agree with the conclusion drawn from Figure 2. 2D shows that fewer seems to show that there are fewer complexes with fewer novel components predicted into Human from Yeast than vice versa, (although it’s hard to make this visual comparison since the axes are not aligned), which to me could be used to argue that the human complexome must already be better annotated than yeast (which most would agree is not the case). At the very least, I think “indicates” is way too strong a word to use here, and that more justification should be added for this conclusion if there is sound reasoning behind it.

Reviewer 2 ·

Basic reporting

The authors previously developed a tool called NetAligner that integrates network alignment with orthology information to predict protein complexes. In this manuscript, the authors argue that NetAligner performs better than protein complex predictions based on orthology information alone. In addition, the authors apply NetAligner to several species including human, yeast, fly, and M. pneumoniae, and predict many new protein complexes in these species. The basic reporting of this manuscript adhere to PeerJ policies and standards.

Experimental design

Methods are clearly described with sufficient information. Although the tool NetAligner was previously developed, this manuscript applies the tool to make new and useful predictions.

Validity of the findings

The authors claim that NetAligner does better than protein complex prediction based on simple orthology mapping. Figure 1 shows that NetAligner is better in precision, but worse in recall. Based on this figure, it cannot be claimed that NetAligner is better than orthology mapping.

In addition to demonstrate that NetAligner is better than orthology mapping, it is essential to show that NetAligner has better precision when recall is controlled, i.e. when NetAligner is comparable to orthology mapping in terms of recall. One simple way of adjusting the recall for orthology mapping is to adjust the criteria for orthology to make it more or less stringent. The recall for NetAligner can be similarly adjusted.

In Figure 3, the authors compare the functional homogeneity of query complexes and predicted complexes by NetAligner. Given that a central theme of the manuscript is to compare NetAligner with orthology mapping, it will be useful to comment on the functional homogeneity of predicted complexes by orthology mapping as well.

Additional comments

I find it interesting that only 55% of all predicted yeast complexes are functionally homogeneous, as predicted from M. pneumoniae complexes. This cannot be due to lack of functional annotations, because yeast has the most comprehensive functional annotations of all eukaryotic species. How do the authors explain this low functional homogeneity? Is this because these predictions are less reliable, or is this because these predictions probe aspects of yeast biology that are less well studied?

In Figure 1B, the legend says that "error bars denote one standard error of the mean". However, these error bars seem too large for standard errors. Do the authors mean standard deviations instead?

In Figure 4, the authors show a few examples of their complex predictions. The query complex in these examples are all fairly simple, involving at most one interaction between two different proteins. Are there good examples where the query complex has at least two interactions between different proteins?

In the Methods section on interactome construction, the authors should specify what kinds of interactions are included. For example, are non-binary interactions included? Are genetic interactions included?

---

## Round 0.2 · accepted · Accept

· Academic Editor

Accept

Both reviewers are satisfied with your revisions and recommend publication.

·

Basic reporting

Looks good. I think the adjustments were helpful.

Experimental design

Looks good. The additional specificity of how the interactions were curated, and especially making the input interactions available for download, resolved my concerns here.

Validity of the findings

Besides the precision-recall concern, all the concerns were well-addressed. On the precision-recall concern, the way the authors chose to change the discussion is somewhat helpful, although it would be much stronger if they could follow one of the approaches I or the other reviewer suggested in order to make a fair comparison of their approach with the simpler orthology-based approach. As it is, there is no way for the reader to be confident that the new method is truly an improvement.

For my part, as the authors added a description of the difficulties with following any of these suggestions, I am comfortable with this resolution.

Reviewer 2 ·

Basic reporting

No Comments.

Experimental design

No Comments.

Validity of the findings

No Comments.

Additional comments

The authors have adequately addressed the reviewer's comments.